# Supramolecular dynamic binary complexes with pH and salt-responsive properties for use in unconventional reservoirs

**Bhargavi Bhat**[1], **Shuhao Liu**[1☉], **Yu-Ting Lin**[1☉], **Martin L. Sentmanat**[1], **Joseph Kwon**[1,2], **Mustafa Akbulut**[1,2,3]*

**1** Artie McFerrin Department of Chemical Engineering, Texas A&M University, College Station, Texas, United States of America, **2** Texas A&M Energy Institute, College Station, Texas, United States of America, **3** Department of Materials Science and Engineering, Texas A&M University, College Station, Texas, United States of America

☉ These authors contributed equally to this work.
* makbulut@tamu.edu

**Data Availability Statement:** All relevant data are within the manuscript and its Supporting Information files.

## Abstract

Hydraulic fracturing of unconventional reservoirs has seen a boom in the last century, as a means to fulfill the growing energy demand in the world. The fracturing fluid used in the process plays a substantial role in determining the results. Hence, several research and development efforts have been geared towards developing more sustainable, efficient, and improved fracturing fluids. Herein, we present a dynamic binary complex (DBC) solution, with potential to be useful in the hydraulic fracturing domain. It has a supramolecular structure formed by the self-assembly of low molecular weight viscosifiers (LMWVs) oleic acid and diethylenetriamine into an elongated entangled network under alkaline conditions. With less than 2 wt% constituents dispersed in aqueous solution, a viscous gel that exhibits high viscosities even under shear was formed. Key features include responsiveness to pH and salinity, and a zero-shear viscosity that could be tuned by a factor of ~280 by changing the pH. Furthermore, its viscous properties were more pronounced in the presence of salt. Sand settling tests revealed its potential to hold up sand particles for extended periods of time. In conclusion, this DBC solution system has potential to be utilized as a smart salt-responsive, pH-switchable hydraulic fracturing fluid.

## Introduction

In a swiftly expanding world, the demand for energy and resources have been steadily increasing. In an effort to keep up with the demand, focus has shifted towards the production of shale throughout the world. This has led to an increased focus on hydraulic fracturing to extract shale gas in unconventional reservoirs [1–3]. It is crucial that the selected fracturing fluid mixture has sufficient viscosity to transport proppant (sand particles) effectively to create fractures in the rock formations [4]. The process itself involves pumping the fracturing slurry at a high

**Funding:** This work was supported by the U.S. Department of Energy, Office of Fossil Energy under award number DE-FE0031778 granted to M. A. The funders had no role in study design, data collection and analysis, decision to publish, or preparation of the manuscript.

**Competing interests:** The authors have declared that no competing interests exist.

pressure to propagate fractures in the reservoir, which serve as conductive paths for shale gas to flow out of the reservoir and into the wellbores [5–7].

Fracturing fluids based on guar gum and its derivatives account for about 90% usage in this application due to their low cost and adequate rheological characteristics [8]. But, the severe damage it can cause to the reservoir (i.e., called formation damage) is a major area of concern [9, 10]. There has been movement towards the use of slickwater to combat some of the issues associated with traditional linear or crosslinked gels [11]. However, its low viscosity and sub-par proppant carrying capability have been considered as a major drawback [12]. Another variety available is foam based fracturing fluids, which are formed by adding pressurized gas, usually $CO_2$ or $N_2$ to water-based or oil-based gels [13–15]. They can contain up to 95% gas and are thus preferred because of their low water requirement, but are limited in application due to complications arising from high costs and logistical issues [16, 17]. Viscoelastic surfactant fluids have been a step towards achieving clean fracturing fluids with internal breaking properties, such that they can be reused [18, 19]. They form elongated wormlike micelles in solution which entangle and form a network to yield increased solution viscosity [20, 21]. Some of their key drawbacks include insolubility under saline conditions and high leak off rates [22, 23]. Another alternative to improve the properties of fracturing fluids is the use of supramolecular complex based fluids, which have received much attention recently [24–29]. It is important to underline that some chemical flooding fluids developed for enhanced oil recovery may also be utilized in some hydraulic fracturing applications [30–32].

Supramolecular complexes consist of dynamic building blocks whose interactions are primarily non covalent and reversible in nature [33]. Some of the interactions that may be taking place between the constituent molecules are: hydrogen bonding, $\pi-\pi$ stacking, metal-ligand coordination, hydrophobic interactions, halogen bonding and electrostatic interactions [34–36]. This feature enables responsiveness to an external stimulus such as pH, temperature, salinity, light, redox activation, etc [37–40]. Furthermore, the weak nature of the forces between the constituents allows for the formation of interesting 3-dimensional structures via a facile self-assembly process [41, 42]. These characteristics make these complexes a prime choice for use in the design of smart materials [35, 43].

In this paper, a novel salt-responsive, pH-switchable dynamic binary complex (DBC) will be presented, which is a special type of supramolecular complex. It is characterized by the use of low molecular weight viscosifiers (LMWVs), at moderate loading in water. The gelation property is dependent on the ability of the LMWV to trap the solvent molecules via capillary forces [44]. This DBC involves intermolecular complexation of oleic acid, which is the major fatty acid present in olive oil, and diethylenetriamine. The aforementioned solution can be switched between low-viscosity and high-viscosity regimes by simply changing the pH. This would be useful in the hydraulic fracturing space since it can be kept viscous while carrying the proppant and its viscosity can then be reduced during the flowback stage. Its viscous properties were pronounced upon the addition of salt to the mixture. Rheological characterization of the solution was done as a function of pH and salt concentration. Furthermore, sand settling tests were also performed to assess its proppant carrying capability. The fact that salt enhances its properties is a major advantage since it opens up the possibility of using seawater, rather than precious freshwater for fracturing applications.

## Materials and methods

### Materials and sample preparation

Oleic Acid (>90%, technical grade) was purchased from Alfa Aesar (Ward Hill, MA, USA). Diethylenetriamine (>98%, for synthesis), Sodium Chloride (>99%, ACS Reagent) and

Polyacrylamide were procured from Sigma Aldrich (St. Louis, MO, USA). Blue colored sand particles were obtained from Ashland Global Specialty Chemicals (Wilmington, DE, USA). The solutions were prepared by mixing oleic acid and diethylenetriamine in distilled water such that the total weight of the two components added up to less than or equal to 2% weight of the solution. Also, sodium chloride was also stirred in some of the mixtures as 1 wt% or 3 wt % of the total solution to introduce effects of ion solvation. To homogenize the solutions, the constituents were stirred in a beaker with a magnetic stirrer for ~2 hours. Both 1 wt% and 2 wt % DBC solutions were made in this manner. The molar ratio of the two components were fixed at 3:1 (oleic acid:diethylenetriamine) to maximize non-covalent interactions between amino groups and carboxylic acid group. After the dispersion of the components, the pH of the mixtures was adjusted by the addition of 10 wt% $H_2SO_4$ or 10 wt% NaOH.

## Rheological measurements

Viscosity measurements were conducted at 23˚C using a rotational rheometer (Haake RS 1, Thermo Fisher Scientific, Waltham, MA, USA) configured with 20 mm diameter titanium parallel plate (PP20 Ti) fixture with a gap setting of 1 mm. The steady shear viscosity curves were determined by sweeping the shear rate logarithmically and stepwise (15 steps) from 0.001 $s^{-1}$ to 100 $s^{-1}$. The shear rate was held constant at each recorded point for 10 seconds while the viscosity was measured. Zero shear viscosities were determined by extrapolating the viscosity versus shear rate curves to near zero values.

For measurement of viscoelastic properties, another rotational rheometer (DHR-2, TA Instruments, New Castle, DE, USA) configured with parallel plate geometry of 40 mm diameter was used. The gap setting employed was 1 mm. Dynamic shear frequency sweep measurements were performed on all of the samples at a strain amplitude that varied from 1–10%, depending on the sample used. In particular, 1% strain amplitude was chosen for pH 11 and pH 12 samples whereas 4% and 10% was chosen for pH 9 and pH 10 respectively. These amplitude values were chosen depending on the limit of linear viscoelasticity of each sample. The angular frequency was swept from 0.1 to 100 rad $s^{-1}$. For each sample that was tested (both viscosity and viscoelastic measurements), three replicate rheological measurements were performed. The mean of these values was used to plot the curves and standard error from the mean was calculated as well. Due to the formation of precipitates below a pH of 8, the range of operation of this study was kept basic (pH 9–12). The salinity of the solution was also varied to examine the effect of salt on viscosity. The total loading of the viscosifiers in solution was 1–2 wt%.

## Sand settling test

To gauge how the dynamic binary complex solutions would hold up proppant, sand settling tests were performed. 7 wt% of blue colored sand particles were dispersed and agitated in vials of dynamic binary complex solution of differing pH and salinity. Settling experiments were conducted at temperatures of 23˚C and 90˚C. For the lower test temperature measurements, high-resolution images were taken after the following intervals: 1 min, 10 min, 1 hr, 2 hr, 4 hr, 8 hr, and 24 hr to see how much of the dispersed sand settled within that time frame. Accounting for the faster settling times at the higher test temperature, images were taken after the following intervals: 20s, 1 min, 2 min, 5 min, 10 min, and 1hr.

## Microscopy analysis

In order to visualize the microstructural changes that take place in the supramolecular solution, images were taken using an optical microscope (Bioryx 200, Arryx Inc, Chicago, IL, USA). Images were captured at pH values of 9 and 11 to understand the formation of

microstructural entanglements and supramolecular behavior that leads to major changes in the rheological properties.

## Results and discussion

### Effect of salinity on viscosity of DBC suspension

The supramolecular complexation of oleic acid and diethylenetriamine resulted in a dispersion with a viscosity changing not only with pH but also with salinity. To better understand the influence of each parameter, we first screened a combinatorial range of pH and salinity in the range of 8 to 12 and 0 wt% to 3 wt%, respectively. We found that the maximum viscosity was obtained at a pH value of 11 and a salinity of 3 wt%. In this section, we describe how the viscosity of such supramolecular complexes changes with respect to salinity at a fixed pH of 11 to elucidate the sole influence of salinity on the system (Fig 1). Intriguingly, the overall viscosity increased as the salt concentration was increased, in the range of 0 wt% to 3 wt%. At a shear rate of $10^{-3}s^{-1}$, there was an about 260-fold increase in the viscosity upon increasing salinity from 0 wt% to 3 wt%. At all salinity values, the supramolecular complex demonstrated a shear-thinning behavior.

It is interesting to delve into why dynamic supramolecular complexation of a triamine and a monocarboxylic acid leads to this behavior. There can be a few reasons behind these trends. First, to date, a wide array of spectroscopic, diffraction and simulation studies have been conducted to acknowledge and interpret the effects of ion solvation in water [45]. Dissolved ions in an aqueous solution have been documented to have a strong effect on the hydrogen bonding network of water and therefore on the viscosity of water [46]. Second, it has been understood

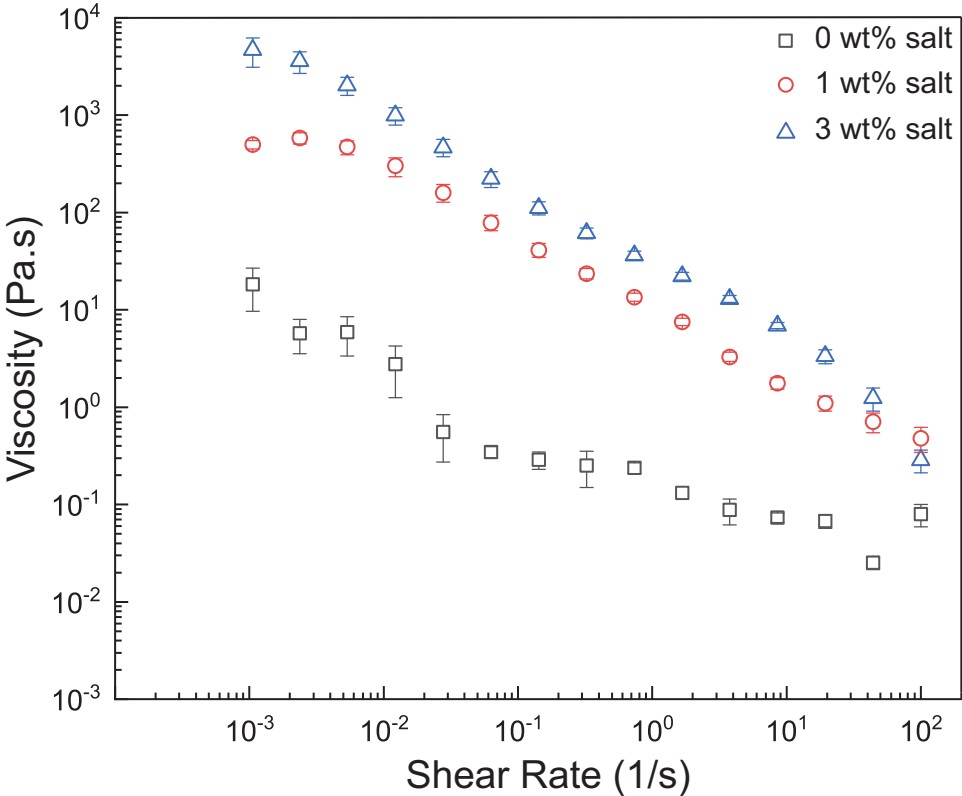

**Fig 1. Effect of salinity on the viscosity of 2 wt% DBC at pH 11.** Upon the addition of salt, the viscosity of the system increased in the salinity range of 0 wt% and 3 wt%. The error bars indicate standard error.

that the presence of solvated ions in the solution decreases the Debye screening length of other charged particles that are present [47, 48]. With reference to this context, the distance up to which the charge on the oleate anions is "felt" in its vicinity decreases when salt is added. Thus, the electrostatic repulsions are screened such that hydrogen bonding interactions between disassociated oleic acid and neutral diethylenetriamine can dominate [49–51].

The use of freshwater for fracturing has been an environmental and sustainability concern for a long time [52]. Most conventional viscoelastic surfactants have the problem of low salt tolerance [53] and hence require freshwater for their preparation. Therefore, a formulation such as the one presented in this paper, which can be prepared using salt water from the ocean would prove to be both ecofriendly and economically viable. In order to gauge the feasibility of this option, the 2 wt% DBC was prepared using simulated sea water, prepared using the major components of a commonly used recipe[54]. Similar trends were observed for the viscosities of these samples which is reported in S1 Fig.

## Effect of pH on viscosity of DBC suspension

After establishing the influence of salinity, we now focus on how suspension pH alters the viscosity of dynamic supramolecular complexes. Fig 2 shows the viscosity trends of supramolecular complexes (2 wt% in water) as a function of pH at a salinity of 1 wt% and 3 wt%. It was found that the viscosity at pH 11 and 12 was about two-orders of magnitude higher than that at pH 9 and 10 for a salinity of 3 wt%, indicating a strong pH-responsive nature of this system. In this case, even at higher shear rates of 100 s$^{-1}$, the viscosity values were in the range of 40–500 cP. While similar trends were also observed at 1 wt% salinity, the adjustability in viscosity with pH was lower, with a 20/40-fold increase upon changing pH from 9 and 10 to 11 and 12. For both salinity values, the supramolecular complex demonstrated a shear-thinning behavior. In particular, log-log plot of viscosity versus shear rate had a straight line at pH 11 and 12, suggesting the applicability of the power-law model for the system. Below pH 9, the supramolecular system exhibited phase separation. Accordingly, the viscosity data was only shown for the pH range of 9 to 12.

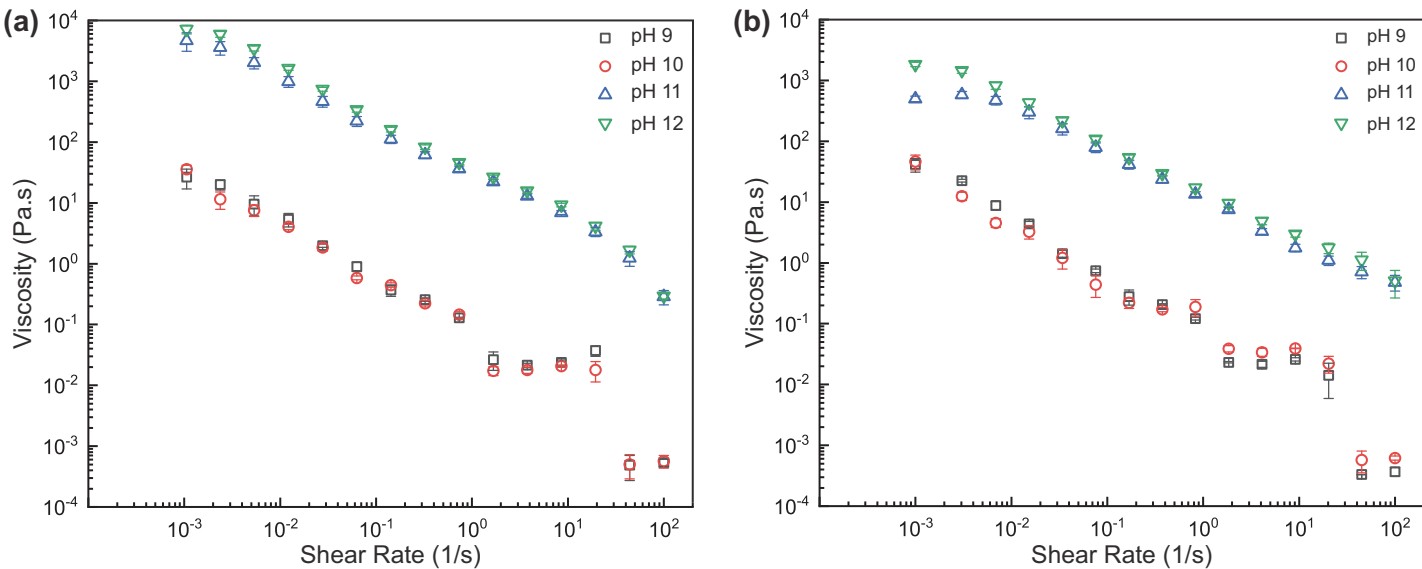

**Fig 2.** (a) Viscosity variation with pH of a 2 wt% DBC solution + 3 wt% NaCl and (b) Viscosity variation with pH of a 2 wt% DBC solution + 1 wt% NaCl. In both figures, the error bars indicate the standard error.

This switchable property can be comprehended by diving deeper into the nature of interactions between the constituent components of the dynamic binary complex. The pKa of oleic acid is around 9.85 [55]. Therefore, above a pH of 9.85, it is completely present in its disassociated form as an anion. Furthermore, as the pH is increased, the diethylenetriamine deprotonates completely and becomes a neutral molecule. Thus arises a situation where hydrogen bonding can take place between oleic acid and diethylenetriamine, such that self-assembly of the constituents into long elongated wormlike structures can occur [56]. The dispersed micelles would undergo uniaxial growth to transition from spherical structures to rod micelles as the salt screens the repulsions [57]. Taking the scission energy of the micelle into consideration, the phenomena here is indicative of a larger value which would propagate growth of the micelles into long entangled structures [57]. This happens even at a relatively low quantity of the constituents. This phenomenon is explained further in section 3.5. The pH tunable material characteristic presents the option of easy cleanup of the reservoir by simply reducing the pH. This would prevent residue formation in the fissures, which has been a leading cause of concern since it reduces the relative permeability of gas [58].

Fig 3 displays further analysis of viscosity versus shear rate data as a function of pH and compares zero shear viscosity of the developed supramolecular complexes with polyacrylamide (PAM), which is a major component of commercial fracturing fluids known as slickwater [12]. It can be seen that the PAM solutions did not exhibit any obvious sensitivity to pH whereas the zero -shear viscosity of the DBC solution increased by a factor as large as ~280 when the pH was increased from 9 to 12. It is also important to underline that for a given solution

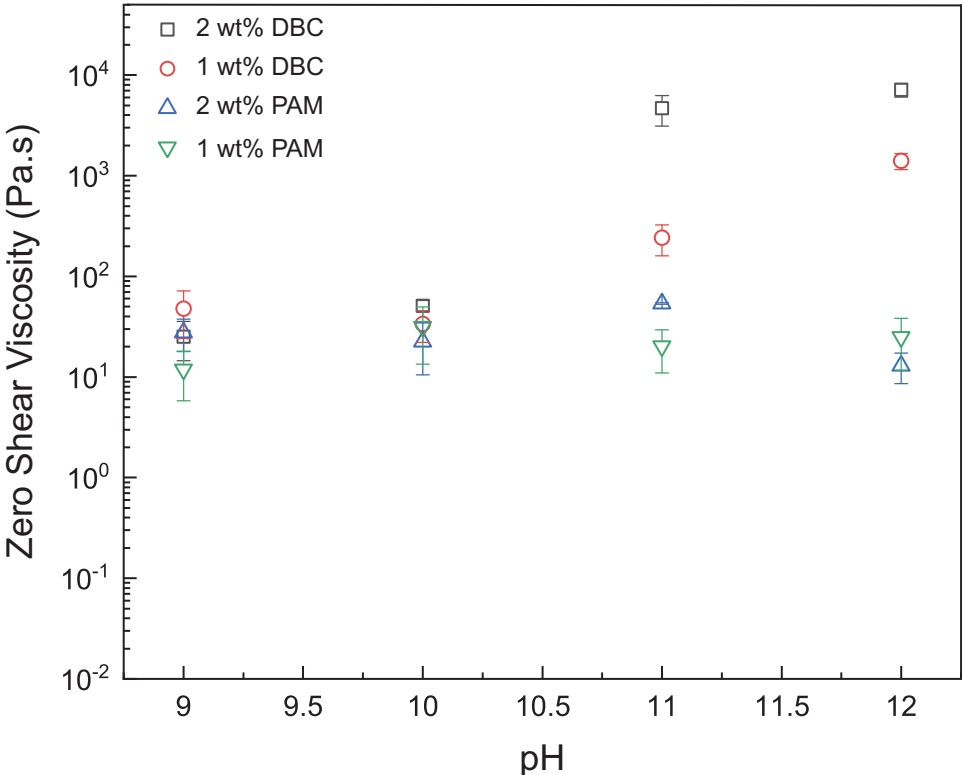

**Fig 3. Comparison of zero shear viscosity for 2wt% and 1wt% polyacrylamide (PAM) solution and 2wt% and 1wt% DBC solution.** All the solutions contain 3 wt% NaCl. pH-tunable nature is observed for the DBC solutions. The error bars indicate standard error.

concentration (1 wt% or 2 wt%), the supramolecular complexed viscosifier had a significantly higher viscosity compared to the PAM solution at pH 11 and 12. Considering that the zero shear viscosity is an important parameter that determines proppant transport [59], one can expect that DBC solution should have a promising potential in hydraulic fracturing applications.

## Viscoelastic characteristics of DBC suspension

In order to further evaluate the rheological dynamics of the DBC solutions at various levels of pH, the storage (G') and loss (G") moduli were measured in the angular frequency range 0.1– 100 rad/s for the 2 wt% DBC+3wt% salt solution (Fig 4). The near constant values of G' and G" at pH 11 and 12, with G'dominating throughout the frequency range of study, are indicative of large characteristic relaxation times that are consistent with the rheological behavior of

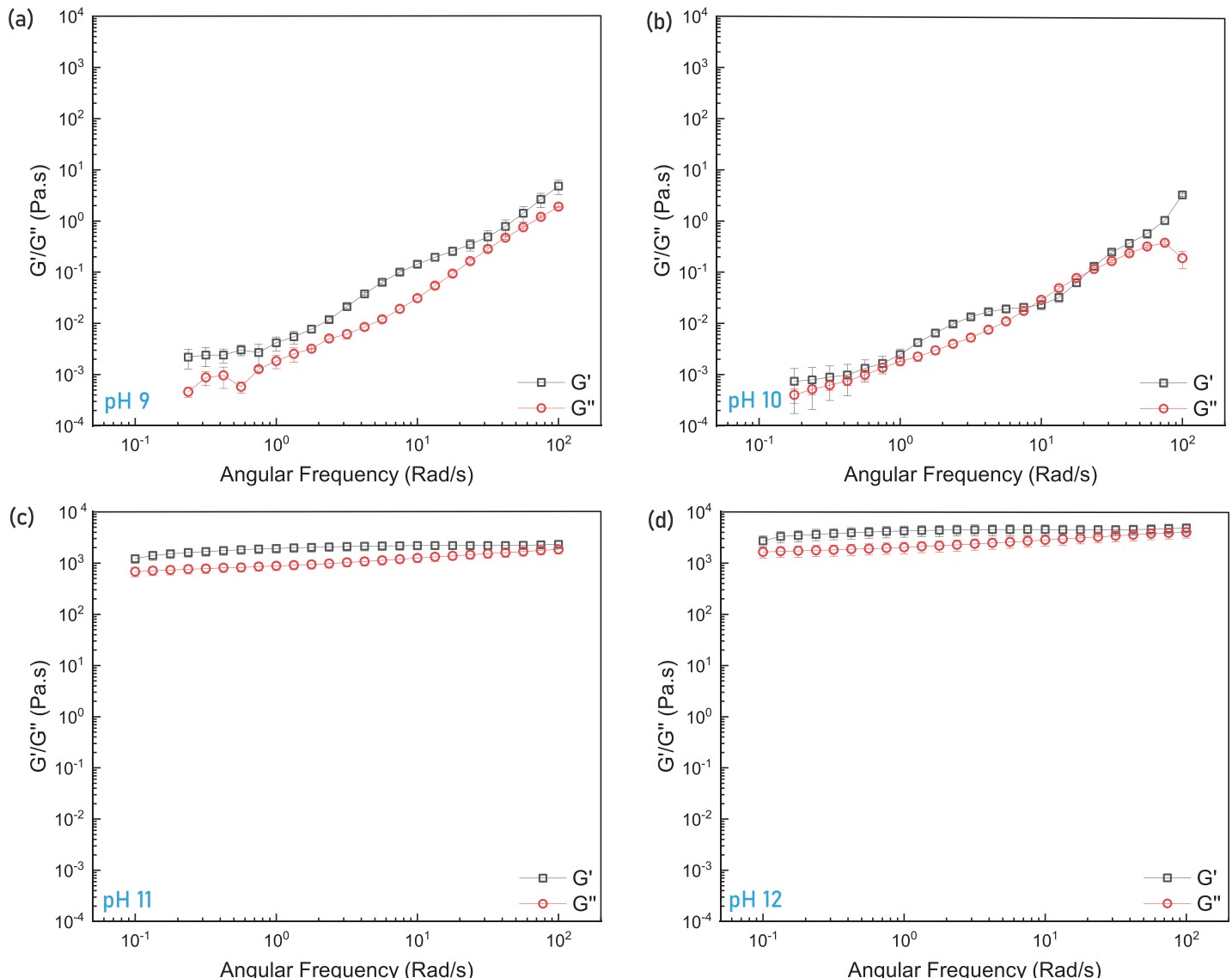

**Fig 4.** Storage (G') and loss (G") moduli of 2wt% DBC+3wt% salt at (a) pH 9, (b) pH 10, (c) pH 11, (d) pH 12. The error bars indicate standard error.

a polymer network or gel [60]. The relaxation time is a measure of the average "lifetime" of the aggregates in the solution, the time after which the system loses its elastic behavior and flows like a fluid [60]. A large relaxation time implies that the supramolecular structure that is viscosifying the solution remains intact and does not disintegrate. This points towards a highly structured complex fluid, similar to the viscoelastic behavior that has previously been observed in crosslinked polymer based fracturing fluids [61]. The predominately elastic behavior exhibited over the entire range of applied frequencies indicates the type of fluid characteristics that would increase the resistance of proppant settling [62]. At a pH level of 9 and 10, both the moduli exhibit a stronger frequency dependence, behavior which would facilitate proppant settling [63].

## Sand settling test

Sand settling test helps present a clear picture of the potential of the DBC as a proppant carrier. The results obtained for both 1 wt% and 2 wt% DBC have been promising. Fig 5 shows images of dispersed sand in DBC solution for 1 wt% DBC+1 wt% salt and 1 wt% DBC+3 wt% salt. Superior sand carrying capability is clearly visible at pH 11 and pH 12 in both cases. Even after 24 hours, the sand stayed dispersed in the matrix and did not settle. This is enough for most hydraulic fracturing applications, which requires sand of varying mesh sizes to be dispersed in the fluid until it settles to the bottom of the fracture [64, 65]. The contrast in the sand settling times was clearly visible for the DBC at pH 9 and pH 10 since it could not hold sand for even a minute at that pH. Hence, we have visual proof of the pronounced pH-sensitive switchable nature. Furthermore, without the presence of NaCl in the solution, the sand settling times were observed to be in the range of a few seconds, allowing us to conclude that the results are enhanced by salinity.

In order to assess its application potential in reservoirs, where the fracturing fluid is exposed to higher temperature gradients, especially during the flowback period [66], the sand settling test was also conducted at an elevated temperature of 90 ˚C. There are some reservoirs that fall within the range of the study conducted, such as the Andarko basins, which reach relatively low temperatures of 149˚F-207 ˚F (65˚C—97˚C) [67]. Another example would be the Barnett shale, whose average temperature is around 150 ˚F (66 ˚C) [68]. The results for 2 wt% DBC+3 wt% salt are presented in Fig 6. Though the overall sand settling time decreased as compared to the ambient temperature case, a pH-switchable behavior was still observed. The samples at pH 12 had a settling time of around 5 minutes in contrast to almost immediate settling at pH 9 and pH 10. The decreased viscosities at this temperature along with the pH tunability could prove to be beneficial during flowback of fluid.

## Microstructural analysis of DBC viscosifiers

With the intention of visually inspecting the microstructure that gives rise to the viscous gellike behavior, optical microscopy was utilized at pH 9 and 11 as shown in Fig 7. An entangled network of fibrous structures was seen in the micrograph for pH 11 whereas circular segregated structures were observed at pH 9. It has previously been reported in literature that in the pH range of 8–10, this system forms giant vesicles in solution and forms a more network-like structure as the pH is increased even more [56]. It is now visually quite conceivable that selfassembly of the constituents into the network like structure is what contributes to the large viscosity values and gel formation. Furthermore, a previous study that highlighted the relationship between proppant suspension behavior and microstructure alludes towards the importance of fibrillar network structure of a gel in its ability to hold up proppant [69].

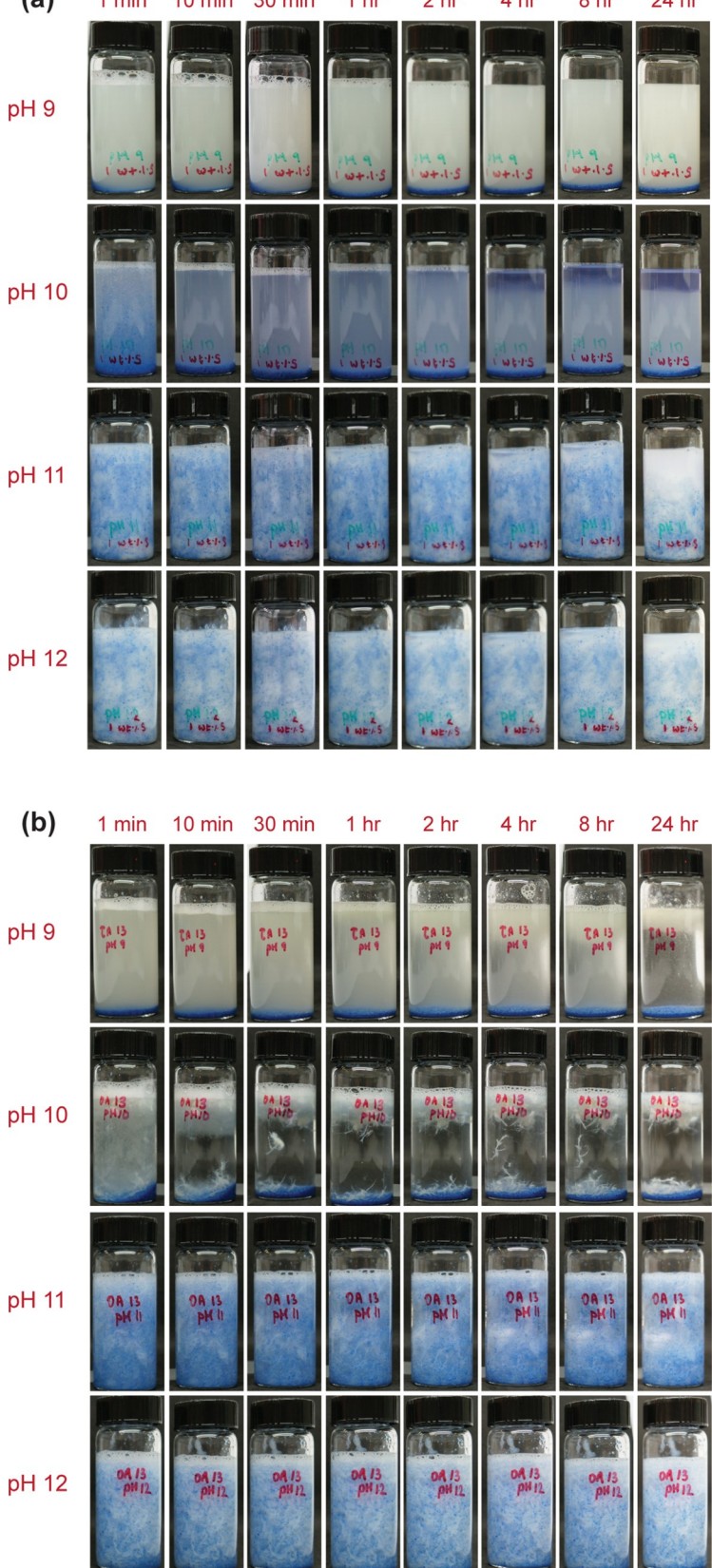

**Fig 5.** Sand settling test for (a) 1 wt% DBC + 1 wt% salt and (b) 1 wt% DBC + 3 wt% salt at room temperature (23˚C).

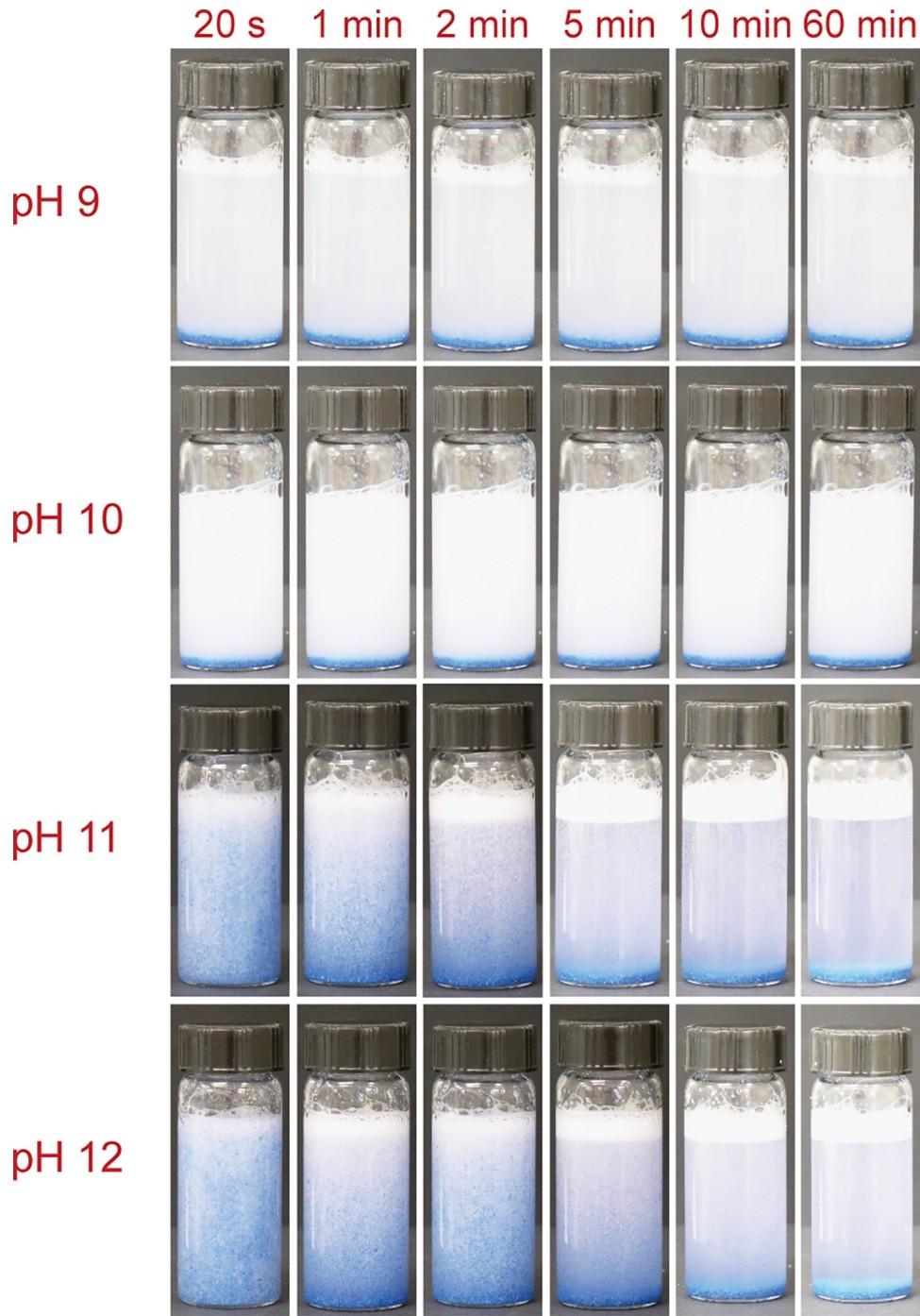

**Fig 6. Images showing the stability of proppant suspension in saline DBC solution (2 wt% DBC + 3 wt% salt) at 90˚C during various time intervals.**

## Application potential

During the process of hydraulic fracturing, it is desirable to have an initially high viscosity during proppant transport. After the fluid injection step is completed and the proppant is deposited in the fractures, the fracturing fluid must attain a lower viscosity to allow flow back and

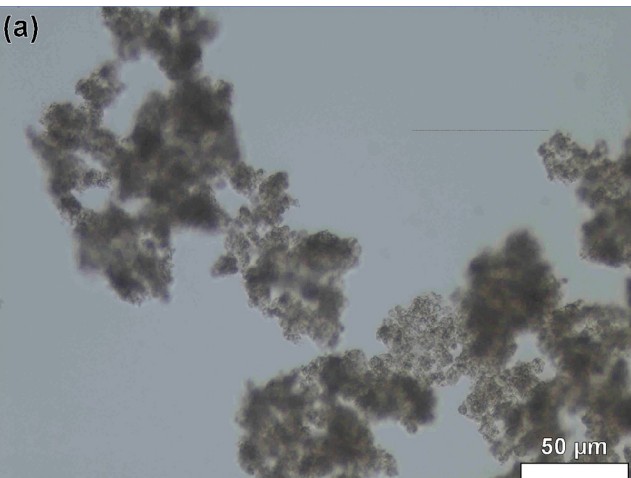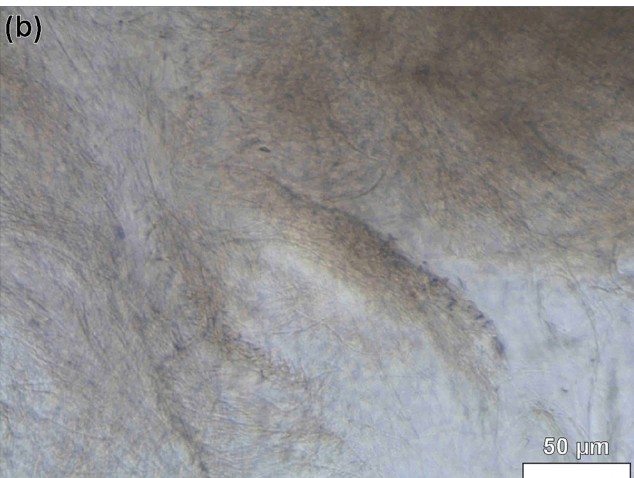

**Fig 7.** Optical microscopy images of supramolecular DBC solutions at (a) pH 9 and (b) pH 11. Scale bar indicates 50 μm.

efficient cleanup [70]. The use of oxidizers and enzymes have sometimes been utilized for this purpose in linear and crosslinked gels, but their incompatibility towards certain fracturing fluids and additives prompted the search for alternatives [71, 72]. The dynamic binary complex presented in this paper, which can be switched down to a lower viscosity by decreasing the pH has the potential to address the gel breaking requirements of industry in a facile manner. Commercially, borate cross-linked fracturing fluids are widely used pH switchable aqueous fracturing fluids [73, 74]. However, they have been reported to have poor stability in the presence of seawater [73]. Our dynamic binary complex system poses the additional advantage of working under saline conditions.

Traditional viscoelastic surfactants usually utilize external conditions in the reservoir such as contact with hydrocarbons or brine in order to break the gel and reduce the viscosity [75, 76] However, there are disadvantages associated with this, especially in dry reservoirs, and it has been noticed that relying on external conditions to break the fluid often creates the need for remedial cleanup [77, 78]. The utilization of an acid as a breaker, as proposed in this paper would have the potential to deliver more consistent results since the addition of acid can be controlled by the operator. It is also possible to consider the use of biodegradable acids like citric acid to fulfill this purpose, increasing the ecological viability of the process.

For hydraulic fracturing applications, the stimuli-responsive nature of DBCs will allow one to precisely control the injection of proppants into fissures and the removal of gelling matrix by dynamical disassembly (Fig 8). The lack of "nanostructured" network and mesh under the decomplexed state will avoid the permeability damage to fissures. Moreover, under extreme shear forces, the DBCs can disassemble into the molecular building blocks rather than covalently fragmenting and scissioning. Then, when the systems reach regions with less shear stress, the building blocks can complex-back into DBCs having high viscosity again. Finally, the favorable interactions between silica and silica-based materials and DBC chains will facilitate the adsorption of DBC on the surfaces/walls of fractures, which will in turn reduce the fluid loss due to leak-off during fracturing step. When the fracturing is done, the adjustment of pH will deprotonate and reprotonate DBC functional groups and will force DBC adsorbates to desorb away from the fracture walls, fully allowing fossil fuels to flow into fractures without any restrictions.

## Shale Gas Recovery

**Fig 8. Illustration of the use of stimuli-responsive dynamic binary complexes as viscosifying agents in shale gas recovery.** The reversibly adjustable viscosity is a key property that can allow a precise control over proppant deposition and fluid flowback.

## Conclusion

This paper presents a novel dynamic binary complex (DBC) whose viscous and viscoelastic properties are highly responsive to pH and salinity. The preparation of the solution involved stirring together oleic acid and diethylenetriamine at a molar ratio of 3:1 (oleic acid: diethylenetriamine). DBC solutions of both 2 wt% and 1 wt% were prepared and both of them exhibited a gel-like behavior under saline conditions at a pH of 11 and above. Their viscosity could be brought down by decreasing the pH, thus making them pH tunable solutions. Viscoelastic tests suggested a complex fluid structure favorable for holding proppant. Additionally, sand settling tests of these solutions exhibited their potential to hold proppant a much longer duration compared to traditional fracturing fluids. Optical micrographs showed the differences in self-assembled structures at pH 9 and pH 11 which were responsible for the divergent

rheological properties. This formulation has the potential to be used as a hydraulic fracturing fluid whose viscosity can be tuned with pH, with the potential to improve proppant transport. An additional advantage lies in its potential to be prepared with seawater, rather than freshwater due to its efficacy in saline conditions.

## Supporting information

**S1 Fig. Viscosity variation with pH of 2 wt% DBC prepared in simulated sea water.** (DOCX)

## Author Contributions

**Conceptualization:** Bhargavi Bhat, Mustafa Akbulut.

**Formal analysis:** Bhargavi Bhat, Mustafa Akbulut.

**Investigation:** Bhargavi Bhat, Shuhao Liu, Yu-Ting Lin.

**Writing – original draft:** Bhargavi Bhat.

**Writing – review & editing:** Shuhao Liu, Yu-Ting Lin, Martin L. Sentmanat, Joseph Kwon, Mustafa Akbulut.

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
