## [Decision Letter · Decision Letter 0]

2 Sep 2021

PONE-D-21-21931

Supramolecular dynamic binary complexes with pH and salt-responsive properties for use in unconventional reservoirs

PLOS ONE

Dear Dr. Akbulut,

Thanks for your submission.  As you see the reviewer is quite positive, but has several minor comments. Please take these into account for your revision. I do not anticipate a need to request a second review from the reviewer, provided the points are taken into account.  Please make modifications to the manuscript clear.

We look forward to receiving your revised manuscript.

Kind regards,

Jay D. Schieber, Ph.D.

Academic Editor

PLOS ONE

Journal Requirements:

This material is based upon work supported by the U.S. Department of Energy, Office of Fossil Energy under Award Number DE-FE0031778.

This work was supported by the U.S. Department of Energy, Office of Fossil Energy under award number DE-FE0031778 granted to M.A. The funders had no role in study design, data collection and analysis, decision to publish, or preparation of the manuscript.

Reviewers' comments:

Reviewer's Responses to Questions

**Comments to the Author**

1. Is the manuscript technically sound, and do the data support the conclusions?

Reviewer #1: Yes

2. Has the statistical analysis been performed appropriately and rigorously? 

Reviewer #1: Yes

3. Have the authors made all data underlying the findings in their manuscript fully available?

Reviewer #1: Yes

4. Is the manuscript presented in an intelligible fashion and written in standard English?

Reviewer #1: Yes

5. Review Comments to the Author

Reviewer #1: Article: Supramolecular dynamic binary complexes with pH and salt-responsive properties for use in unconventional reservoirs

Overall Rating: publish with minor revision required

This article by Bhargavi Bhat et al. presents rheological measurements on a dynamic binary complex (DBC) solution with PH and salt-responsive properties with the potential to be used in hydraulic fracturing of unconventional reservoirs. The rheological properties of samples depend on PH and salinity. They also have done sand settling experiments to check the possibility of holding the sand particles up.

The manuscript is rather well written, and the results are also correctly discussed. I think that this study is worth interest for the PLOS ONE after some modifications. The following comments may help the authors to improve their manuscript:

1. Line 88: .  ,

2. Line 104: in your plots, the minimum shear rate is 10^-4 instead of 0.001 1/s.

3. Line 110: in cone- plate geometry, it is not possible to adjust the gap. How do you adjust the gap to 1 mm?

4. Line 110-112: “Dynamic shear frequency sweep measurements were performed on all of the samples at a strain amplitude of 1-10% over an angular frequency range from 0.1 to 100 rad s‒1.”

Strain amplitude is constant in oscillatory frequency sweep tests. It is not correct to have a region of 1-10% for strain amplitude. Would you please clarify it?

5. Line 119: Why do you choose 7wt% of sand particles for the settling tests? Any calculation according to the real situation? Or randomly was chosen?

6. Line 143: shear rate’s unit is 1/s, not HZ!

7. Line 144: increasing the viscosity at a shear rate of 10^-3 s^-1, is not 30-fold by increasing salinity from 0wt% to 3wt%. Please check the amount of viscosity from the plot.

8. Line 223: What is the amount of the imposed shear strain/stress in the angular frequency sweep (fig 4)?

9. How the error bars are calculated in Fig. 1? Do you repeat the test several times?

10. Could you provide higher resolution images in Fig 5?

Good luck!

6. PLOS authors have the option to publish the peer review history of their article (what does this mean?). If published, this will include your full peer review and any attached files.

Reviewer #1: No

---

## [Author Response · Author response to Decision Letter 0]

14 Sep 2021

Dear Dr. Scheiber,

We thank the reviewer for judiciously reading our manuscript and for their constructive comments and suggestions on our manuscript, “Supramolecular dynamic binary complexes with pH and salt-responsive properties for use in unconventional reservoirs.” We have carefully considered these comments/suggestions. In light of the feedback we have received, we have revised the article accordingly. Below are our responses to the comments (Changes are marked with red in the revised main text).

Editorial Comments:

The funding information has been removed from the acknowledgements section of the manuscript. We are fine with the current funding statement, and it does not have to be updated. The ORCID ID of the corresponding author has been linked. The reference list was judiciously reviewed, and it was ensured that there were no retracted articles. However, reference 67 (now 70) was noticed to be incomplete and that has been corrected. 3 new references regarding supramolecular systems to adequately capture the state-of-art in this area have been added to the manuscript as well. For Figure 5, we have uploaded a higher resolution image (<10 MB size limit). However, if the 10 MB can be waived, we can upload even higher resolution image that can satisfy the reviewer’s request. Kindly use the high-resolution image if possible.

Reviewer Comments

1. Line 88:  ,

Thank you for pointing out this typographical error. The period in this line has been replaced with a comma, as it is more appropriate.

2. Line 104: in your plots, the minimum shear rate is 10^-4 instead of 0.001 1/s.

We appreciate this comment and see the issue with the axes of the figures. In order to address this, we have changed the figures (Fig1 and Fig2) to ensure that the labelled axes start at a shear rate of 10^-3 (0.001 1/s).

3. Line 110: in cone- plate geometry, it is not possible to adjust the gap. How do you adjust the gap to 1 mm?

Thank you for this observation. We realized that there is an error in this sentence. In fact, we used a 40 mm parallel plate, Peltier plate steel for which gap adjustment is possible. This line has been corrected in the manuscript.

4. Line 110-112: “Dynamic shear frequency sweep measurements were performed on all of the samples at a strain amplitude of 1-10% over an angular frequency range from 0.1 to 100 rad s‒1.” Strain amplitude is constant in oscillatory frequency sweep tests. It is not correct to have a region of 1-10% for strain amplitude. Would you please clarify it?

Thank you for your comments. We meant to say that the strain amplitude chosen was 1% for certain samples (pH 11 and 12) and greater values for the others (4%/10%). This is because a larger amplitude seemed necessary for the instrument to effectively detect the modulus values of less viscous samples. Furthermore, it was possible to have a larger strain amplitude for pH 9 and 10 samples because they have a larger limit of linear viscoelasticity. The above statement has been modified in the manuscript to elucidate this point. 

5. Line 119: Why do you choose 7wt% of sand particles for the settling tests? Any calculation according to the real situation? Or randomly was chosen?

Thank you for your question. In literature, proppant settling tests have been conducted with a wide variety of concentration of sand, as low as 5 wt% [1]. Sometimes, the approach has been to just use a single particle of sand and see how fast it settles down after travelling through the fluid [2]. This particular value of 7wt% was chosen as a good starting point which would allow us to gain a visual picture of how well sand is being captured by the matrix of the fluid.

6. Line 143: shear rates unit is 1/s, not HZ!

This has been corrected in the updated manuscript

7. Line 144: increasing the viscosity at a shear rate of 10^-3 s^-1, is not 30-fold by increasing salinity from 0wt% to 3wt%. Please check the amount of viscosity from the plot.

Thank you for this correction. A more accurate statement would be that there is a ~260 fold increase in viscosity at 10^(-3) s^(-1) upon changing the salinity from 0 wt% to 3 wt%. This correction has been made in the main text.

8. Line 223: What is the amount of the imposed shear strain/stress in the angular frequency sweep (fig 4)?

This has been addressed in question 4. The corrected statement in the manuscript provides this information in a better fashion under the ‘Materials and Methods’ section

9. How the error bars are calculated in Fig. 1? Do you repeat the test several times?

Thank you for your question. Each measurement was performed 3 times for statistical analysis. The error bars represent the standard error from the mean of the 3 measurements. We recognize this has not been mentioned clearly in the text. Hence, we have added an explanation in the materials and methods section.

10. Could you provide higher resolution images in Fig 5?

Due to the image size limit, Figure 5 had to be compressed. Some resolution was lost because of this. A slightly better version that is under 10 MB has been uploaded. The link for the original high resolution file has been shared with the PLOS One as well.

1. Ahmad FA, Miskimins JL. Proppant Transport and Behavior in Horizontal Wellbores Using Low Viscosity Fluids. 2019. doi:10.2118/194379-MS

2. Shah SN. Proppant Settling Correlations for Non-Newtonian Fluids Under Static and Dynamic Conditions. Soc Pet Eng J. 1982;22: 164–170. doi:10.2118/9330-PA

Sincerely,

---

## [Editor Report · Decision Letter 1]

17 Nov 2021

Supramolecular dynamic binary complexes with pH and salt-responsive properties for use in unconventional reservoirs

PONE-D-21-21931R1

Dear Dr. Akbulut,

We’re pleased to inform you that your manuscript has been judged scientifically suitable for publication and will be formally accepted for publication once it meets all outstanding technical requirements.

Kind regards,

Jay D. Schieber, Ph.D.

Academic Editor

PLOS ONE

---

## [Editor Report · Acceptance letter]

24 Nov 2021

PONE-D-21-21931R1 

Supramolecular dynamic binary complexes with pH and salt-responsive properties for use in unconventional reservoirs 

Dear Dr. Akbulut:

I'm pleased to inform you that your manuscript has been deemed suitable for publication in PLOS ONE. Congratulations! Your manuscript is now with our production department. 

Kind regards, 

on behalf of

Prof. Jay D. Schieber 

Academic Editor

PLOS ONE